# Ocean Acidification and Aquacultured Seaweeds: Progress and Knowledge Gaps

**Tan Hengjie** [1], **Simon Kumar Das** [1], **Nur Farah Ain Zainee** [2], **Raja Yana** [3] and **Mohammad Rozaimi** [1,*]

1   Department of Earth Sciences and Environment, Faculty of Science and Technology,
    Universiti Kebangsaan Malaysia, Bangi 43600, Malaysia
2   Department of Biological Sciences and Biotechnology, Faculty of Science and Technology,
    Universiti Kebangsaan Malaysia, Bangi 43600, Malaysia
3   Department of Fisheries Malaysia, No. 30, Persiaran Perdana Precinct 4, Putrajaya 62628, Malaysia
*   Correspondence: mdrozaimi@ukm.edu.my

**Abstract:** This systematic review aimed to synthesise the existing studies regarding the effects of ocean acidification (OA) on seaweed aquaculture. Ocean acidification scenarios may increase the productivity of aquacultured seaweeds, but this depends on species-specific tolerance ranges. Conversely, seaweed productivity may be reduced, with ensuing economic losses. We specifically addressed questions on: how aquacultured seaweeds acclimatise with an increase in oceanic $CO_2$; the effects of OA on photosynthetic rates and nutrient uptake; and the knowledge gaps in mitigation measures for seaweed farming in OA environments. Articles were searched by using Google Scholar, followed by Scopus and Web of Science databases, limiting the publications from 2001 to 2022. Our review revealed that, among all the OA-related studies on macroalgae, only a relatively small proportion ($n < 85$) have examined the physiological responses of aquacultured seaweeds. However, it is generally agreed that these seaweeds cannot acclimatise when critical biological systems are compromised. The existing knowledge gaps regarding mitigation approaches are unbalanced and have overly focused on monitoring and cultivation methods. Future work should emphasise effective and implementable actions against OA while linking the physiological changes of aquacultured seaweeds with production costs and profits.

**Keywords:** acclimatisation; macroalgae; physiological response; productivity





## 1. Introduction

The global seaweed aquaculture industry has contributed to the production of various downstream and upstream products such as food, biopolymers, cosmetics, nutraceuticals, bioenergy compounds, and pharmaceuticals [1]. The production of seaweed-based biofuel as an alternative to fossil fuel [2] has managed to reduce up to 1500 tons $CO_2$ km$^{-2}$ year$^{-1}$ when compared to emissions from fossil fuels [3]. Among its other functions, the open ocean aquaculture of seaweeds provides shoreline protection from storms and waves [3,4]. Seaweed production can also help to reduce ocean eutrophication by absorbing nutrients required for seaweed growth [5]. With a wide distribution of biomass at the global level, Seaweed Aquaculture Beds (SABs) have the potential to at least act as a temporary carbon sink to mitigate the immediate effects of climate change [6]. This is due to the seaweed's capacity for carbon assimilation and accumulation, and $CO_2$ sequestration in a relatively short period [4,7]. On the other hand, there is evidence indicating that certain naturally growing seaweeds have the capacity for carbon sequestration and accumulation, which can be exported and buried in deep sea regions [8,9]. However, with the elevation in atmospheric $CO_2$, ocean acidification (OA), as one of the impacts of climate change, will negatively affect entire marine systems. Although this is a globally pressing matter, the discourse on the potential ecological or economic impacts of seaweed production is still limited.

Seaweeds are used in many different industries after they are harvested. As such, seaweed cultivation has grown from constituting 45.9% of global mariculture production in 2004 to 51.3% in 2018. In terms of monetary value, this represents 24.2% of the associated worldwide profits, worth US$ 28.1 billion [10,11]. The combined production output of China, Indonesia, Japan, the Republic of Korea, the Philippines, and Malaysia contributed up to 99.5% of global seaweed mariculture production in 2018 [11–13]. This region led the world's seaweed production, with an estimated at 32.4 million tonnes in 2018, compared to a threefold lower production in 2000 (10.5 million tonnes; [14]). Global seaweed cultivation then increased to 35.1 million tonnes in 2020 for food and non-food uses [15]. The highest output in terms of seaweeds by quantity farmed within temperate and tropical regions are: *Eucheuma* (also known as gusô, 10.2 million tons in 2015), Japanese kelp (8 million tons), *Gracilaria* (sare, 3.9 million tons), *Undaria* (wakame, 2.3 million tons), *Kappaphycus* (elkhorn sea moss, 1.8 million tons), and *Porphyra* (nori, 1.2 million tons) [1,12,16]. From the temperate region, seaweed production in Norway reached a sales value of EUR 74,000 in 2017, while studies have predicted that annual sales will rise further to EUR 4 billion by 2050 [17].

Up to 95.6% of all seaweed used by humans is aquacultured to ensure sustainability in terms of supply and to prevent the overexploitation of the natural population [10]. The remainder is harvested from naturally growing beds. Brown seaweeds, which comprise nearly 64% of the farmed production, are harvested for a variety of uses, including human nutrition and alginate extraction. They are used in various sectors including the medical industry, textile printing, and paper coating [1,6,18,19]. Other seaweeds such as *Chondrus crispus* (Irish moss), *Kappaphycus*, and *Eucheuma* are used as gelling material, emulsifiers, and stabilisers in the pharmaceutical, cosmetic, and food processing industries [1,13,18].

Environmental and biotic stressors can negatively affect the growth and productivity of macroalgae including aquacultured seaweeds [20]. The sources of these stressors vary from marine pollution [21,22], disease outbreaks [23–25], epiphytic infestation, algal parasites [25], periodic storms [26], and global warming [23]. Rising sea-levels due to global warming may cause shifts in shoreline morphology, with the subsequent effect of lowered productivity output through changes in seaweed distributional patterns [20]. Such changes may have more pronounced effects on naturally growing seaweed beds that would be harvested for seaweed industries rather than those in floating cultures [20]. The impact of sea-level changes on floating cultures might not be obvious but changes in pH can still be considered one of the factors which affect their physiological responses [27] (Table 1).

Knowledge on seaweed physiology, especially concerning how environmental stressors affect the productivity of aquacultured seaweed, is clearly essential to ensure the success of seaweed farming [1]. Ocean acidification (OA) is an inherent stress factor for optimal seaweed growth. Ocean acidification, or a reduction in seawater pH, stems from atmospheric $CO_2$ dissolving in seawater, eventually acidifying the water via the production of carbonic acid ($H_2CO_3$). This then dissociates into bicarbonate ions ($HCO^{3-}$) and protons ($H^+$) [28]. The seawater chemistry is altered when hydrogen ions [$H^+$] increase and the concentration of carbonate ions ($CaCO_3$) are reduced. This eventually causes a decrease in oceanic pH and leads to OA conditions [29,30]. According to the Representative Concentration Pathway (RCP) 6.0, atmospheric $CO_2$ emission will rise to 700 ppm by the year 2100. Ocean pH is expected to decline by 0.3–0.5 units towards the end of the century, with an estimated corresponding increase of 100–150% in [$H^+$] [28,31–33]. Thus, carbonate saturation states are predicted to decline by approximately 45% by 2100 [34–37].

Studies indicate that the metabolic rates and population growths of marine organisms can endure increases in oceanic $CO_2$ up to their physiological threshold limits because of their ability to acclimatise within an optimum range of pH values [38,39]. However, a failure to acclimatise would put those species at risk of mortality [38,40]. In this regard, although studies regarding the impacts of OA to marine fauna or fisheries aquaculture have been conducted [38,41,42], knowledge gaps remain surrounding the impacts of OA on seaweed aquaculture. While these limits are more discernible for calcifying macroalgae, the tolerance of fleshy macroalgae to OA is still unclear, i.e., whether such environmental

changes will positively or negatively affect their productivity [43,44]. The physiological responses of non-calcifying seaweed towards OA are species-specific and inconsistent at different developmental stages [45], mostly due to different carbon-uptake strategies [43,46]. Furthermore, the interactive effects of OA and other environmental variables such as temperature complicates any definitive prediction with regard to the exact impacts of OA effects on fleshy seaweed [45].

In this review, we discuss how the increase in dissolved $CO_2$ with pH variation will affect the physiological responses of aquacultured seaweeds. In particular, we focused on directions towards answering the following questions: (1) How do aquacultured seaweeds acclimatise to an increase in oceanic $CO_2$? (2) What are the effects of OA on the photosynthetic rates and nutrient uptakes of aquacultured seaweeds? And: (3) What are the knowledge gaps in the mitigation strategies for future seaweed aquaculture considering an ocean-acidified environment?

## 2. Materials and Methods

This review follows the guidelines provided by Preferred Reporting Items for Systematic Reviews and Meta-Analyses (PRISMA) and the methodology is presented in a PRISMA flow scheme (Figure 1) [47].

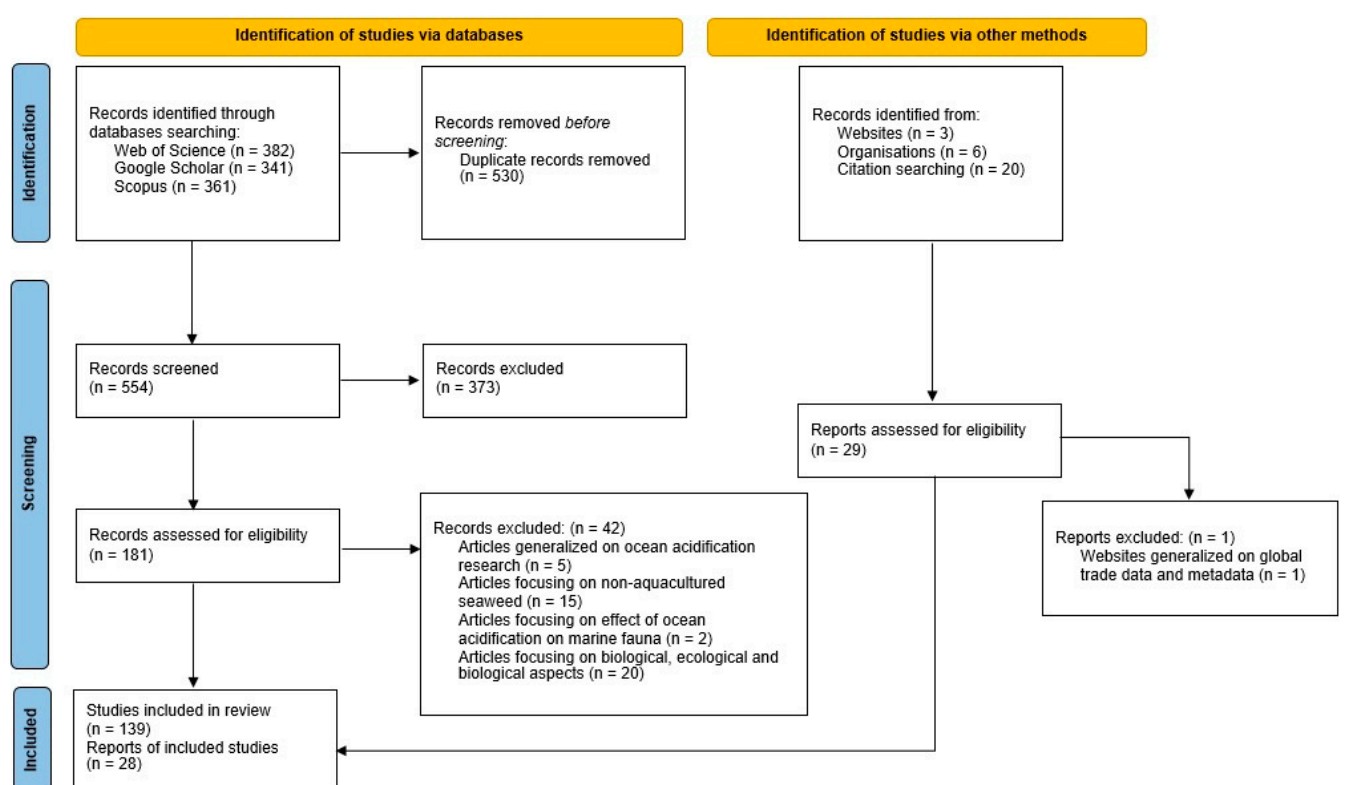

**Figure 1.** Flow diagram of the systematic review.

### 2.1. Resource Identification

We broadly approached the review of the literature based on the methodology suggested by Snyder (2019) [48]. We first defined the main purpose and research question using the following scopes: (1) the specific seaweed species aquacultured in (2) scenarios of OA conditions. A stepwise search strategy was started by limiting specific keywords, which were selected based on the two scopes. We trawled Google Scholar, Scopus, and Web of Science as the main literature databases. This step would cover peer-reviewed articles that are accessible through the public domain. The keyword searches in these databases included terms such as "ocean acidification," "elevation of dissolved $CO_2$," "seaweed aquaculture," "seaweed productivity," "photosynthetic rate", and "physiological

response." To achieve a wide coverage in terms of sources for a comprehensive literature search, we added the keyword input by choosing related terms such as "seaweed adaptation", "seaweed acclimatization", "climate change", "global warming", "macroalgae", and "seaweed industry". We applied additional techniques for advanced-phrase searches by combining strings developed and strengthened using Boolean operators (i.e., "AND") in Google Scholar and Scopus. The strings "Ocean acidification" AND "Carbon dioxide" AND "Seaweed" AND "Aquaculture" AND "Physiological responses" returned 702 articles published between 2001 and 2021 (Figure 1). Similarly, when trawling the Web of Science database, the combination of search strings in the field tag to obtain topic hits (TS) were: (TS = (Ocean acidification)) AND (TS = (Seaweed)) AND (TS = (Physiological Response)). This search returned 382 results for papers published between 2012 and 2021. The timeframe is consistent with the methodology for scoping systematic reviews [48], and specifically for this study, and was stipulated to ensure that no species were overlooked. Furthermore, the emphasis on this timeframe for article selection was to minimise the duplication of subject species and to ensure that the published information provided was up to date. To avoid the duplication of articles in the literature search, all articles obtained were imported into a reference manager (Mendeley), as suggested by Li et al. [49].

### 2.2. Scoping and Screening

Keeping in mind the main scope of the research question (i.e., surrounding aquacultured seaweeds and OA), the search hits were first screened based on the titles and abstracts of the papers to determine their suitability and relevance. During the article trawling, a limited number of non-peer reviewed technical papers and database reports were included in the screening, but only if these articles were highly cited (>100 times), such as those reporting on the economic and management aspects of seaweed aquaculture [22], as well as selected Food and Agriculture Organization (FAO) [12,21,50,51] and Intergovernmental Panel on Climate Change reports [36,37,50,52]. Studies of seaweed species in aquaculture, farmed in both tropical and temperate regions, were shortlisted. Their relevance was further refined to consolidate the literature that directly related to the environmental variables of OA. To ensure a sufficiency of sources obtained for this review, we did not limit our literature search based on studies about the physiological effects of aquacultured seaweeds due to environmental stressors such as global warming and climate change. Rather, we also focused on search results related to the potential of aquacultured seaweeds as a natural mitigation tool for OA. This is in addition to other articles that highlighted any economic valuation—positive or negative—related to OA on the seaweed aquaculture industry. As a result, we made certain that our collection included narratives of the relationship between dissolved $CO_2$ elevation and aquacultured seaweed species, based primarily on their physiological responses (Table 1), followed by any subsequent impacts on industry-level aquaculture production. Any studies of aquacultured seaweed species that dealt with biological, ecological, and biogeochemical aspects that were not directly related to OA were excluded during the screening. Articles in languages other than English were also excluded (Figure 1).

### 2.3. Article Analysis

To provide a more insightful perspective to address question (3), a further analysis of the identified articles relating to the mitigation of the effects of OA on aquacultured seaweeds was performed. A first screening was carried out through keyword searches for terms that allude to mitigation approaches, such as site buffering, site selection, framework, and monitoring [53,54]. This was carried out to determine the main keywords that would be analysed, which would then lead to ranking the most discussed mitigation-related strategies in the literature. We found 131 papers which addressed or discussed mitigation to varying extents while focusing on mitigation as the main thematic area. The overarching theme relates to effective global measures involving synergy between policies, regulatory frameworks, and strategy implementation [14,55,56]. To ensure the results were unbiased [48],

studies were analysed to determine if the search findings met these specific criteria, in that: (i) the approaches mentioned had been conducted; (ii) the current state of the strategy implementation; or (iii) challenges faced or efforts needed to further develop the mitigation measures. The keywords were searched manually throughout the articles based on occurrence frequency in the main text body of the articles. The total number of specific word hits from each article were recorded accordingly while excluding the same keywords if these appeared in the reference list (see Supplementary Data Table S1). We found a limited number of research articles with the same keywords (Supplementary Data Table S1). Certain keywords appeared less frequently in research articles compared to technical reports, although a similar phrasal context was applied by the authors of the respective articles (Table 2). Based on the article analysis and keyword search, 35 articles were identified as being the most relevant. Up to 10 keywords were finally shortlisted, which were either mentioned as the exact word/phrase in the article or discussed in a similar context. The specific term of 'management' was not considered as a keyword but as a root word since the definition could be further clarified by using the other identified keywords. These keywords and the context used was summarised to avoid an overlapping analysis (Table 2). The amount of specific word hits was summed and converted into percentages representing the proportion of each keyword that appeared out of the overall search (Supplementary Data Table S1) before ranking the 10 components (based on the shortlisted keywords) of mitigation approaches. Lastly, the ranked data was visualised graphically through a radar chart (Figure 2). This chart aims to list the top findings based on the queried article set. Its use allows for a focus on assessing the overall inherent issues and potential mitigation strategies for OA impacts on seaweed aquaculture. The visual presentation and holistic approach of the radar chart summarises the results and highlights the knowledge gaps that would require further studies.

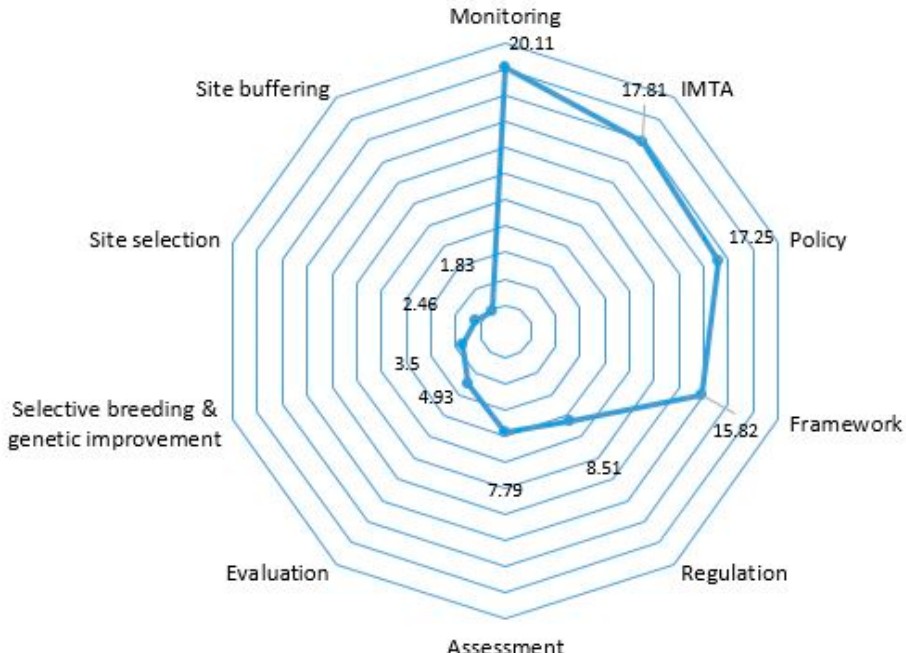

**Figure 2.** Radar chart that illustrates the existing knowledge gaps in relation to the main thematic areas for mitigating the effects of OA on seaweed aquaculture.

After the compilation was performed, we then identified, analysed, and interpreted any presence of consistent trends in the texts of the shortlisted literature to highlight the most relevant subtopics for further discussion [48]. We discussed these subtopics based on the headings of the following sections (see below). These sections are consistent with the themes of seaweed physiology, environmental stressors, seaweed cultivation, and aquaculture management, based on expert opinions and interpretations in the studies in

question, in the context of evaluating the prospects of seaweed production in OA scenarios. The synthesis of studies published in the past 32 years that linked aquacultured seaweeds with OA conditions was then conceptualised (Figure 3) and tabulated (Table 1).

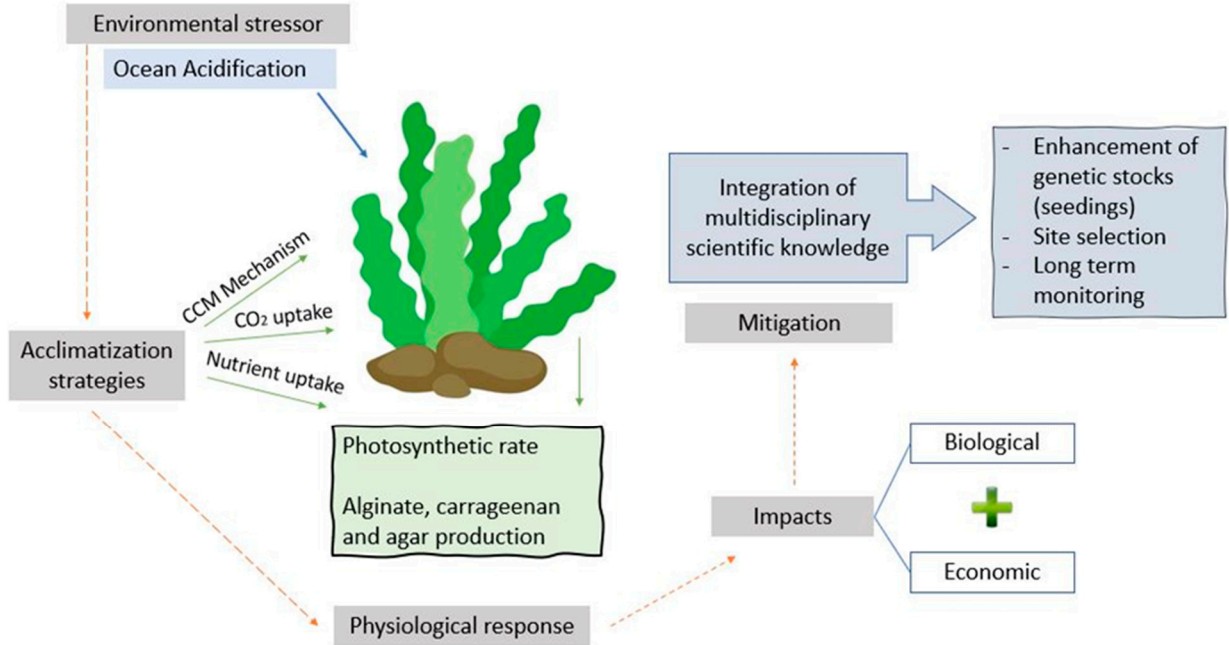

**Figure 3.** A conceptual diagram, which consists of various aquaculture aspects aimed at sustaining the productivity of aquacultured seaweeds under OA scenarios.

**Table 1.** The impact of OA on seaweed species in temperate and tropical regions. The CCMs of types 1 to 3 are described in Section 3.

| Region | Species | CCM Type | Variable | Main Findings | References |
|---|---|---|---|---|---|
| Temperate | *Alaria esculenta* | 3 | Photosynthetic activity, biochemical composition (lipid content), enzymatic activities (eCA) | Increase in growth, lipid content, and photosynthetic efficiency ($F_v/F_m$) under elevated $CO_2$, with lower photon requirements; enzymes are not sensitive to changes in $CO_2$. | [57–60] |
| | *Saccharina japonica* | 1, 2, 3 | Iodine accumulation, photosynthetic efficiency, photosynthetic oxygen, germination | Tissue growth enhanced under lower pH with a simultaneous increase in iodine accumulation; inhibition of photosynthetic rate is relatively higher under lower pH, and the photosynthetic efficiency ($F_v/F_m$) is not much affected; reduction in meiospore germination and reproduction rate. | [10,51,61–66] |
| | *Saccharina latissima* | 2 and 3 | Photosynthetic acclimation, pigment composition | Photosynthesis and growth rates are negatively affected; CCMs are deactivated; optimal temperature for growth is 5–15 °C; no effects on biochemical composition. | [17,34,57,67–70] |
| | *Undaria pinnatifida* | 2 and 3 | Gametophyte development | No significant impact on meiospore germination but increase in germling growth rates and gametophyte sizes when seawater pH is reduced from 8.40 to 7.20; rates of net photosynthesis (NPS) of gametophytes and juvenile sporophytes start to decrease when pH drops from 7.20 to 5.5. | [1,10,12,65,71–73] |
| | *Pyropia* sp. | 1 and 2 | Photosynthetic rate, growth rate | Increase in growth and nutrient uptake; growth of thalli enhanced by 30% at pH of 6 and 7; photosynthetic rate increases when pH drops from 7 to 6, but photosynthesis and respiration rate decrease at pH of 4 and 5; tissue death when low pH conditions are prolonged. | [1,4,74–80] |

**Table 1.** *Cont.*

| Region | Species | CCM Type | Variable | Main Findings | References |
|---|---|---|---|---|---|
| | *Gracilaria* sp. | 2 and 3 | Growth rate, photosynthesis, photosynthetic inorganic carbon uptake, iodine accumulation | Growth rate increases through carbon uptake; enhanced carbon/nitrogen ration; photosynthetic pigments remain unchanged; increase in photosynthetic acclimation; increase in iodine accumulation under elevated dissolved $CO_2$. | [1,4,12,16,61,81–85] |
| | *Chondrus crispus* | 2 and 3 | Photosynthetic rate | Able to acclimatise when oceanic pH decreases, and photosynthetic rate is maintained; carbon fixation rate is highest at pH of 7. | [1,18,29,86–88] |
| | *Sargassum fusiforme* | 1, 2, 3 | Growth rate, nitrogen assimilation, photorespiration | Photosynthetic rate is maintained under increase in $CO_2$ since the species is able to tolerate pH declines with enhanced relative growths; biomass increase was associated with nitrogen assimilation within tissues. | [12,27,39,89,90] |
| | *Macrocystis pyrifera* | 1, 2, 3 | Germination rate, gametophyte development, iodine accumulation, growth rate, photosynthetic rate | Meiospore germination, gametophyte development, and spore production and recruitment negatively affected in acidified conditions; iodine accumulation slightly increases, and tissue growth exhibited under elevated $pCO_2$; no changes in growth and photosynthetic rate but increased uptake of $CO_2$. | [18,61,90–94] |
| | *Sargassum vulgare* | 1, 2, 3 | Alginate content, polysaccharides content, bioactivity (antibacterial activity, antifungal activity), carbohydrate availability, antioxidant capacity, enzymatic activities, photosynthetic rates | This species is able to acclimatise to low pH conditions of 6–6.7; secondary metabolites are lower; bioactive properties grow naturally in acidified conditions; alginate content higher in acidified conditions; increase in dissolved $CO_2$ results in increased bioactivity, antioxidant capacity, enzyme activity, photosynthetic rate, and polysaccharide content. | [95–99] |
| | *Porphyra* sp. | 2 and 3 | Growth rate | Increase in growth. | [16,94] |
| | *Ulva rigida* | 2 and 3 | Growth rate and assimilation (carbon and nitrogen), HCO3-utilisation, photosynthetic rate, dark respiration rate, soluble protein content, inactivation of CCMs, nitrogen metabolism | Increase in growth rate and nutrient assimilation especially for carbon and nitrogen under acidified conditions; photosynthetic rate, dark respiration, and soluble protein reduced with increased dissolved $CO_2$; photosynthesis process is negatively affected due to the inactivation of CCMs. | [99–101] |
| | *Fucus vesiculosus* | 2 and 3 | Growth, nutritional quality, carbon and nitrogen content, fertility | Reduction in growth and C:N ratio; no changes in other elemental compositions; increase in $pCO_2$ alter temporal development of fertility, according to the changes in temperature seasonally. | [45,101–104] |
| Tropical | *Gracilaria* sp. | 2 and 3 | Growth rate, photosynthesis, gephotosynthetic inorganic carbon uptake, iodine accumulation | Growth rate increases through carbon uptake; no significant effect on maximum relative electron transport rates (rETRmax); increase in iodine accumulation under elevated dissolved $CO_2$. | [1,4,12,82,105–107] |
| | *Undaria pinnatifida* | 2 and 3 | Photosynthesis, gametophyte development, germling growth rate | No significant impact on meiospore germination; increase in germling growth rate, and gametophyte size when seawater pH is reduced from 8.40 to 7.20; rates of net photosynthesis of gametophytes and juvenile sporophytes decrease when pH drops from 7.20 to 5.5. | [12,72,73,108] |
| | *Kappaphycus alvarezii* | 1 and 3 | Daily growth rate (DGR), photosynthesis | DGR decreases at pH 6 due to the low availability of photosynthetic carbon sources in low pH conditions; decrease in efficiency of $CO_2$ accumulation. | [1,12,13,18,93,109] |
| | *Pyropia* sp. | 1 and 2 | Net photosynthesis, growth rate, respiration | Increased growth and nutrient uptake; growth of thalli enhanced by 30% at pH of 6 and 7; photosynthetic rate increases when pH drops from 7 to 6; photosynthesis and respiration rate decrease at pH of 4 and 5; thalli death in prolonged low pH conditions. | [4,74–76,79,80,110–113] |
| | *Eucheuma* sp. | 2 and 3 | Photosynthetic rate | Increase in photosynthetic rate when oceanic pH decreases below 8. | [1,18,94] |

**Table 1.** *Cont.*

| Region | Species | CCM Type | Variable | Main Findings | References |
|---|---|---|---|---|---|
| | *Caulerpa lentillifera* | 3 | Carbon absorption rate | Increase in growth through carbon uptake. | [111,114,115] |
| | *Hypnea* spp. | 3 | Growth rate, maximum quantum yield, chlorophyll a content, antioxidant activity | Decrease in growth rate, maximum quantum yield (fv/fm), and chlorophyll a content; increase in antioxidant activity. | [116–122] |
| | *Gelidium* spp. | 3 | Growth rate, carbohydrate content | Decrease in growth rate; no significant changes in carbohydrate content; reduction in species richness. | [87,123–127] |

**Table 2.** Definition of the terms used in the radar chart, with the selection of words based on the applied contextual meaning to avoid duplication in occurrence counts and their use being overlooked during the keyword search.

| Keyword | Definition in Context | References |
|---|---|---|
| Policy | The related aims to: mitigate $CO_2$ emission to the atmosphere; inform decision-making at local, regional, and national levels in order to integrate into global goals; manage areas that are used for seaweed aquaculture. | [8,128] [7,14,16,56,129] |
| Framework | The provisions for: policy and integrated planning that requires more experimental and innovative practices at different authoritative levels (local, state, or federal jurisdictions); legal frameworks, which refer to guidelines in the setting up and management of seaweed aquaculture; management of fisheries resources and aquaculture governance; conservation and the sustainable use of aquatic living resources. | [16,56] [8,13,54] [14,56,130] |
| Regulation | The provisions to: adequately manage the resources of coastal aquaculture, including seaweed cultivation; coupled with an appropriate monitoring and law enforcement system while banning unsustainable practices. | [13,128] [14,16] |
| Monitoring | Efforts to: measure the local environmental and spatial variability in carbonate chemistry within coastal areas or aquaculture farms; track long-term environmental changes through a combination of efforts by various stakeholders globally; incorporating a Fisheries and Resources Monitoring System (FIRMS) in seaweed aquaculture; improve transparency in fisheries and aquaculture stock and production records. | [50,53,54,131] [17,34,128] [14,56,132] |
| Evaluation | The appraisal of: the application and performance of aquaculture systems that involve several authorities to ensure ecosystem sustainability; the interaction with existing resources that are characteristic of coastal areas and suitable for different types of farming. | [4,14,54,131,133] [13,17,128] |
| Assessment | The inclusion of: managerial tools for quantifying the risks and benefits associated with seaweed aquaculture; diagnosing the current status of stocks in fisheries and aquaculture; Environmental Impact Assessment; | [8,54] [14,34,133] [25,128] |
| Site selection | Related efforts to: choose optimal sites for aquaculture activities within each environment; match seaweed species with specific cultivation techniques; include licensing approvals from authorities. | [4,53,133–135] [13] [50] |
| IMTA | A method of aquaculture that: consist of species components from different trophic levels and serving different ecosystem functions; increases the biomass production and sustainability of aquaculture; mitigates environmental problems caused by specific forms of fed aquaculture | [13,53,130,133] [17,54,130,132] [131] |
| Site buffering | Related efforts to: buffer seawater pH and carbonate chemistry; quantify the ability of seaweeds to buffer the impacts of climate change including OA. | [53] [3,8,135] |
| Selective breeding and genetic improvement | A method of aquaculture that is used: to cultivate seaweeds for specific traits to enhance production and resilience under conditions of projected climate change; to obtain a culture stock that has increased tolerance against the impacts of OA through strain development. | [4,13,14,53,54] [34,136] |

## 3. Ocean Acidification and Seaweeds' Photosynthetic Rates and Nutrient Uptakes

When seaweeds with different photosynthetic capacities are exposed to increasing oceanic $CO_2$ concentrations, biochemical and molecular changes cause varying carbon uptake capacities [39]. This has various implications in terms of alterations in the physiological responses of aquacultured seaweeds (Table 1). Seaweeds that solely obtain $CO_2$ through diffusive uptake are usually known as non-carbon concentrating mechanism (non-CCM) species [137]. These species will exhibit increases in growth and photosynthetic rates under

elevated $CO_2$, with an eventual increase in species abundance instead of diversity [138]. Despite the availability of more dissolved $CO_2$ as a result of OA, and the supply of $CO_2$ as the primary C source is no longer limited, it is worth noting that $CO_2$ diffuses into water at a slower rate than it does in air [27,139]. Most of the fleshy macroalgae in aquaculture are carbon concentrating mechanism (CCM) species. These seaweeds require an adjustment in terms of their kinetic mechanisms to utilise inorganic carbon such as bicarbonate ions to facilitate $CO_2$ delivery to the ribulose-1,5-bisphosphate carboxylase/oxygenase (RubisCO) enzyme for carbon fixation [4]. Nonetheless, different CCM species will respond distinctively towards OA in order to acclimatise and survive because not all have the same affinity towards DIC [46,140].

According to Giordano et al. (2005) [141], there are three types of CCMs: (1) CCMs based on $C_4$ mechanisms and crassulacean acid metabolism (CAM); (2) CCMs based on the active transport of DIC; and (3) CCMs based on changes in $CO_2$ concentration in compartments adjacent to RubisCO, where $HCO^{3-}$ is mostly found in the chloroplast stroma and is converted to $CO_2$ when it crosses to the thylakoid. *Macrocystis pyrifera*, for example, is a type of fleshy macroalgae that uses both $CO_2$ and $HCO^{3-}$ for photosynthesis and growth [91]. Those species with a high affinity for DIC will either exhibit increases in their metabolic processes or will not be obviously affected [140]. In this regard, $HCO^{3-}$ is dehydrated extracellularly by external carbonic anhydrase (CAext), then converted to $CO_2$ by internal carbonic anhydrase (CAint) to be used by CCM seaweeds, or, alternatively, it is accumulated within an internal inorganic carbon pool [39,91]. This mechanism has been described in *Chondrus crispus* and *Sargassum fusiforme* [39,86,96] as well as in other aquacultured seaweeds that use $HCO^{3-}$, including *Alaria esculenta*, *S. fusiforme*, *M. pyrifera*, *C. crispus*, and *Saccharina latissima* [91,142]. In contrast, CCM seaweeds with a low affinity for DIC will benefit under elevated $CO_2$ [137,140].

Aquacultured seaweeds would therefore acclimatise to elevated $CO_2$ levels through various carbon uptake strategies [3,27], which can be determined by the $\delta^{13}C$ value of seaweed tissues. CCM seaweeds typically have $\delta^{13}C$ values ranging from $-30‰$ to $-10‰$. This range, however, varies with changes in $pCO_2$ in the marine environment [137], indicating that species attempt to acclimatise despite the fluctuating $pCO_2$ [34,96]. Species such as *S. fusiforme* can tolerate pH drops while maintaining appreciable photosynthetic rates. Subjected to increasing $CO_2$ levels, this species responds positively based on enhanced relative growth rates and increased biomass associated with significant nitrogen assimilation within its tissues [39,89]. Other studies have shown that *Kappaphycus alvarezii* and *Sargassum vulgare* were able to acclimatise in low-pH conditions (pH 6 and pH 6.7, respectively) [74], while *M. pyrifera* exhibited no changes in growth and photosynthetic rates under elevated oceanic $CO_2$ because it has the ability to utilise both $CO_2$ and $HCO^{3-}$ [91]. Other economically important seaweeds such as *Hypnea pseudomusciformis*, *Porphyra yezoensis*, *Gracilaria chilensis*, *G. changii*, and *G. lemaneiformis* exhibited increased growth rates with an increase in dissolved $CO_2$ concentrations [4,46,81,106,117,143,144]. Hence, the growth of seaweeds partly depends on their $HCO^{3-}$ uptake and utilisation capacity [96], and the impact on their photosynthetic rate is generally species-specific [27]. Certain seaweeds have the potential to mitigate OA through carbon sequestration [3,135,145] due to their respective physiological changes under pH variation, thus leading to high biomass production [7], as was clearly observed for fast-growing fleshy macroalgae (Table 1).

In contrast, other aquacultured seaweeds, notably red seaweeds such as *Gracilaria tenuistipitata*, *Porphyra leucosticta*, and *P. linearis*, experience decreased growth rates in high $pCO_2$ concentrations due to a reduction in photosynthetic capacity, invariably leading to a decline in photosynthetic efficiency [46,82,146–148]. Decreased growth rates are also possible because CCM seaweeds are sensitive to increased $H^+$ concentrations during OA, which can disrupt cellular homeostasis [137]. Such effects have been studied in the case of the changes in *M. pyrifera* meiospore germination (Table 1) under extreme OA at a mean pH of 7.60 [92]. However, questions arise as to the precise timeframe during which the seaweed's physiology remains unchanged or in a positively balanced state despite the

increase in $p$CO$_2$. In a simulated situation of acidified conditions, the photosynthetic rate of *P. linearis* increased but only lasted for approximately 16 h [147]. The low-pH level in an experimental environment resulted in elevated respiration rates and the consumption of accumulated photosynthate, eventually causing a decline in growth rates, especially at a pH of 6.0 [147]. A similar situation was reported for *Porphyra haitanensis*, whereby its thallus growth increased significantly by 30% when the pH decreased by 1–2 units. Lowering pH levels further results in a significant drop in growth rates until the eventual death of its thalli [74]. Additionally, low-pH growth conditions can severely impact alginate and carrageenan production [149,150]. These observations imply that elevations in dissolved CO$_2$ levels can potentially reduce seaweed productivity.

Not all seaweeds undergo the same consequence in terms of changes in growth rate. This is because the rate of nutrient uptake influences the growth rate of seaweeds under elevated CO$_2$ conditions. An increase in $p$CO$_2$ increased *Ulva rigida*'s nitrogen assimilation rate, which in turn increased its growth rate [82,100,151]. *Sargassum fusiforme* has been shown to have higher NO$^{3-}$ uptake rates when exposed to elevated CO$_2$ levels [39,152]. In similar experimental conditions, there were no changes in the uptake of inorganic nitrogen by *M. pyrifera* despite an increase in CO$_2$ [152]. Other factors that influence the growth rate of seaweeds under elevated CO$_2$ include limitations in nutrient availability in the water column due to the frequent increase in species biomass locally or the stratification that is induced by ocean warming [153]. This occurs through the alteration of ocean carbonate chemistry, which is mainly attributed to the increase in ocean temperature with the increase in CO$_2$ concentrations [4,154,155]. Stratification hinders nutrient supply to primary producers from deep to surface waters because of the formation of layers that prevent normal water mixing such as during upwelling processes. Such effects have been reported in the Southern Baltic Sea and the Basque coast of Spain, where *Fucus vesiculosus* and *Gelidium corneum* experienced lowered growth rates due to nutrient limitations caused directly by ocean stratification [4,156–158], while *G. corneum* in the Cantabrian Sea became less productive due to a reduction in nitrate availability [157].

## 4. Potential Mitigation Strategies for Ocean Acidification Impacts on Seaweed Aquaculture

The high production volume of aquacultured seaweed demonstrates industry demands and a significant contribution in terms of farming revenue [159]. The difference between the cost (capital, maintenance, material inputs, and labour) and the income generated, on the other hand, determines the economic sustainability and cost-effectiveness of seaweed cultivation [3,22]. Indeed, if OA has an impact on seaweed production, the market's value chain in terms of supply and demand will be disrupted, resulting in price volatility and declining profit margins in various processing industries and global marketplaces [22,160]. Therefore, a clear motivation to mitigate the effect of OA on seaweed aquaculture revolves around maintaining profitable production output in the circumstances of an ever-changing global climate. Mitigation strategies must be expanded at the regional level [53] and should serve as fundamental frameworks for strengthening policy implementation and to ensure the respective standards for proper seaweed cultivation practice and effective management are delivered [54]. Unfortunately, an overarching management plan that wholly addresses OA, specifically in seaweed aquaculture, is still lacking in many countries [54].

The governance management for seaweed aquaculture—in the flow of policy, framework, regulation, monitoring, followed by assessment and evaluation [14]—ideally combines multiple aspects of mitigation strategies as the essential elements to counteract OA. These key elements would further incorporate the managerial tools for long term implementation, which include site selection, advanced aquaculture techniques, such as integrated multi-trophic aquaculture (IMTA—see below), site buffering, selective breeding, and genetic improvement (Figure 2). These same factors have been emphasised in multiple articles, which provide indirect leads to a more sustainable seaweed aquaculture. Based

on the broad framework established, regulations should be developed in accordance with mutual goals such as the Sustainable Development Goals (SDGs), which may include technical guidelines to ensure the long-term growth of the seaweed aquaculture industry [14,55]. It is also recommended to identify and, if necessary, establish authoritative bodies with a functional role at the national level. Their responsibilities should include ensuring the sustainability of aquaculture practice while adhering to existing regulatory policies [129,161]. Based on this approach, carbon tax implementation can be expanded globally to manage $CO_2$ release in order to reduce greenhouse gas emissions despite a rapidly growing and developing economy [162–164].

## 5. Multi-Disciplinary Approach to Mitigate the Ocean Acidification Impacts on Seaweed Aquaculture

To counteract the consequences caused by OA, the mitigation steps should ideally be approached in concert. The identification of optimal farming techniques may be a viable first step. Different seaweed aquaculture systems, such as a recirculating land-based system, must be further developed, either through open-water systems or tank cultivation [1,160]. The farming method of choice is primarily determined by the farm's scalable capacity and how to maximise productivity based on local environmental factors [20]. In an open-water cultivation, proper site selection and spatial planning are the key elements to ensure the effectiveness of seaweed aquaculture under elevated $CO_2$ scenarios [3,6,13,136]. In Malaysia, farmers rely on the monoline method as the main farming technique, which has increased production from 60 thousand tons in 2006 to 261 thousand tons since 2015 [12]. Open-water cultivation may be replaced by land-based seaweed farming. Although the former is not new [165], more research is required to improve cultivation techniques and ensure nutrient uptake efficiency in vitro. Certain seaweeds grew well in the presence of elevated $CO_2$, but as density increased, nutrient availability became limited due to competition for uptake [132]. Therefore, certain land-based seaweed farming approaches have been implemented through the concept of integrated aquaculture, notably Integrated Multi-trophic Aquaculture (IMTA), to ensure the supply of nutrients required to support sustainability regarding seaweed growth [16]. In general, IMTA, which consists of complementary ecosystems functions within a single aquaculture system, is known as one of the mitigation strategies for marine aquaculture during OA [53,166]. With the co-cultivation of seaweed as a key component, the whole system would lessen the impact of OA through $CO_2$ buffering while maintaining the possibility of further management applications, such as seaweed-based bioremediation [16,53,159,167]. This would ameliorate the effects of increased $CO_2$ while also creating a favourable environment for seaweed cultivation.

Sustainable seaweed aquaculture stresses proper monitoring regardless of the production technology. This can be improved with a long-term monitoring programme using networks such as the Global Ocean Acidification Observing Network (GOA-ON) [168] or the more regional Southeast Asian Global Ocean Observing System (SEAGOOS) [169]. The variation of water parameters within each culture should be monitored and controlled to provide optimal conditions for seaweed cultivation [13,134,170,171] using devices that are regularly calibrated to ensure data accuracy [52]. Choosing cultivars with specific traits that exhibit higher growth capacity, thermal tolerance, and disease resistance brings us one step closer to having a comprehensive mitigation strategy [134,136,172,173]. This strategy can be further leveraged through genetic improvement using hybridisation technology, which is a means of modifying and developing existing cultivar strains for higher biomass yields without limiting the choice of aquacultured seaweed species [134]. Concerning the seaweeds selected for a cultivation system, species with high DIC affinity could be prioritised to optimise $CO_2$ or $HCO_3^{-}$ uptake capacity [106,152]. *Gracilaria* may be one of the choice species to extract organic and inorganic components since it is efficient at assimilating ammonia, phosphate, and DIC [106,151].

Risk assessments based on an adaptive framework should be prioritised as part of the steps to address potential problems, such as threats from climate change [8,174]. The

Ecosystem Risk Assessment (ERA) is one viable approach for identifying hazards from adverse events and their consequences for a specific organism. The quantification of pre-impact levels and qualitative studies should be considered based on the projected effects of OA, followed by the design and evaluation of precautionary methods [175]. Indeed, more efforts to conduct risk assessments for marine plants classified by the International Union for the Conservation of Nature (IUCN), particularly macroalgae, are required in all coastal regions [56]. This is necessary to reduce the selection of farmed seaweed species that may exhibit negative growth rates in stressful and disturbed marine environments [56].

Finally, the findings from a broad-based approach will provide information on how to consolidate existing policies while ensuring the sustainability of seaweed aquaculture [175]. While anticipating the ripple effects of declining seaweed production, which may result in economic losses, it is critical to develop a systematic and practical plan to overcome future challenges in seaweed aquaculture. A conceptual framework was outlined (Figure 3), which includes various aspects of seaweed aquaculture geared toward sustaining productivity despite the constraints in OA scenarios.

## 6. Conclusions

Seaweed aquaculture has the potential to reduce $CO_2$ emissions while also supporting ecosystem services through $CO_2$ sequestration; however, elevated $CO_2$ and OA are likely to have an impact on seaweed production. To mitigate the negative effects of severe OA on aquacultured seaweed, a comprehensive mitigation plan with adequate monitoring is required. Seaweeds, as the largest group of aquacultured species with high productivity by volume, may be affected by regional and global changes in biomass yield. However, the responses of aquacultured seaweeds to OA vary by species, as evidenced by changes in physiological mechanisms such as the photosynthetic rate and nutrient uptake, which affect seaweed productivity. At the same time, the acclimatisation of aquacultured seaweeds to elevated oceanic $CO_2$ depends on their carbon uptake strategies, while kinetic mechanism adjustments would further determine changes in the photosynthetic rate in each species. If OA has a negative effect on seaweed cultivation, the extent of its impact on seaweed production must be quantified, and because industry profitability is determined by seaweed price and operating expenses, monetary loss can be calculated precisely. More studies are needed to quantify the effects caused by OA on the economy. This includes establishing a link between physiological changes in seaweed and industrial productivity in terms of production costs and potential revenues. As a result, multiple mitigation strategies approached from various angles should be implemented to mitigate the effects of OA on aquacultured seaweed. The emphasis should be on addressing existing knowledge gaps in mitigation approaches, which are still imbalanced and overly skewed toward monitoring- and IMTA-centric efforts. This entails multidisciplinary approaches developed through synergy among various stakeholders—from researchers to aquaculture farmers and policymakers—for a more holistic seaweed aquaculture system that incorporates key mitigation tools. In short, the combined effects of OA on biological and economic factors necessitate the implementation of a more collaborative mitigation strategy that incorporates the various multidisciplinary aspects of OA and seaweed production.

**Supplementary Materials:** The following supporting information can be downloaded at: https://www.mdpi.com/article/10.3390/jmse11010078/s1, Table S1: Summary of the number of occurrences of the main keywords during the article screening that relate to mitigation strategies of seaweed aquaculture in ocean acidification scenarios.

**Author Contributions:** Conceptualisation, T.H. and M.R.; methodology, M.R., N.F.A.Z. and T.H.; validation, M.R., R.Y., S.K.D. and N.F.A.Z.; formal analysis, T.H., M.R. and R.Y.; investigation, T.H. and N.F.A.Z.; resources, S.K.D., R.Y. and M.R.; data curation, T.H. and M.R.; writing—original draft preparation, T.H. and M.R.; writing—review and editing, all authors; visualisation, T.H., N.F.A.Z. and M.R.; supervision, M.R., N.F.A.Z. and S.K.D.; project administration, M.R.; funding acquisition, M.R. All authors have read and agreed to the published version of the manuscript.

**Funding:** This research was funded by Universiti Kebangsaan Malaysia, grant number DIP-2021-021.

**Institutional Review Board Statement:** Not applicable.

**Informed Consent Statement:** Not applicable.

**Data Availability Statement:** Not applicable.

**Conflicts of Interest:** The authors declare no conflict of interest. The funders had no role in the design of the study; in the collection, analyses, or interpretation of data; in the writing of the manuscript; or in the decision to publish the results.

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
