# Peer review of "Ocean Acidification and Aquacultured Seaweeds: Progress and Knowledge Gaps"

_jmse, doi:10.3390/jmse11010078_

Round 1
Reviewer 1 Report
As indicated by the title, this review addresses the current knowledge of ocean acidification effects on seaweeds used in aquaculture. The topic can be expected to be of interest to many readers, as summaries of related knowledge for seaweed species relevant in aquaculture are rare.
The authors describe in much detail their research strategy. What they do not say - but what gets apparent if one checks the literature list - is that this review is somehow a review of reviews: It covers only some primary literature or descriptions of case studies and it largely builds on other reviews or book chapters. A more frequent citation of original studies would be preferred, as it would give credibility to those researchers who actually once provided the original information and it would often allow for more detailed information.
I found the use of grammar in this manuscript often problematic. The references between sub-sentences are often unclear or possibly incorrectly constructed, which leads to misleading or even incomprehensible statements. A thorough linguistic revision would be necessary in any case. The usual reference to the involvement of a native speaker would not automatically be helpful in this case, because it would have to be a native speaker who could also grasp the meaning of the text in terms of content and check its correctness. I give some examples at the end.
I would have expected from this review an overview on the carbon uptake mechanisms and carbon concentrating mechanisms in the different species that are listed in table 1 and I strongly suggest to complement the table with such information, which would be elemental to assess the sensitivity of different species to ocean acidification. I also found the first paragraph of section 3 - that is dedicated to this aspect - not very well structured. It would be better to (1) explain CO2 uptake by diffusion, then (2) explain the different CCMs and (3) address knowledge about their distribution among taxa.
Some detailed points below, note that the list is not complete:
Title and throughout the text: I am not a native speaker, but I do not think the verb "to aquaculture" actually exists...
l. 14 may be reduced, not may reduce
l.36 what are "seaweed aquaculture beds (SABs) with a wide distribution of biomass globally"? Why would they (?) or their global distribution (?) have potential to act as a temporary carbon sink?
l. 47/48 Sentence fragmented and incomprehensive
L. 54-57 What is production of output of seaweeds? In any case, production should be singular, not plural...
L.58-59 Significant growth since when?
L. 61 not 95,6 % of all seaweed is "aquacultured", but 95,6% of all seaweed used by humans is "aquacultured" (see above about "to aquaculture")
L63/64 Most brown seaweeds that are cultivated are actually cultivated for human nutrition, not for alginate production...
l.73/75 A negative effect of sea-level-rise on sea-based offshore aquaculture is not easy to understand, given that this takes usually place on floating infrastructructure, please explain...
L. 94 & 223 Different carbon uptakes strategies should be explained in this paper, not just mentioned...
L. 228 why would CCM seaweeds have a low affinity to DIC...you probably rather mean non-CCM seaweeds?
L264/265 would certain seaweeds mitigate OA due to their high biomass or not rather their physiological ability, that permits them to build up high biomass?
L. 269/270 Why would CCM species be particularly sensitive to low pH
L 274 Do you mean the increase lasted for 16 h or photosynthesis alltogether lasted only for 16 h
L. 345/347 Sentence fragmented
L. 375 are more disease resistant
L377 which means to
L379 for a cultivation system
L383-385 Sentence fragmented
L390 what are IUCN categories
L391-393 Not clear
L402 Seaweeds are the largest group of species in aquaculture in terms of what? Volume, species diversity, value?
Author Response
REVIEWER 1
- Comment #1.1
Comments to the Author: As indicated by the title, this review addresses the current knowledge of ocean acidification effects on seaweeds used in aquaculture. The topic can be expected to be of interest to many readers, as summaries of related knowledge for seaweed species relevant in aquaculture are rare.
Response: Thank you for the feedback and your time in reviewing the manuscript.
- Comment #1.2
The authors describe in much detail their research strategy. What they do not say - but what gets apparent if one checks the literature list - is that this review is somehow a review of reviews: It covers only some primary literature or descriptions of case studies and it largely builds on other reviews or book chapters. A more frequent citation of original studies would be preferred, as it would give credibility to those researchers who actually once provided the original information and it would often allow for more detailed information.
Response: To clarify, in the submitted manuscript, 94 of the 167 total citations in the references were from original studies or primary research articles. The remainder consisted of review articles, technical reports, books, and websites, which we incorporated to present a more comprehensive standpoint from different aspects, and to prevent bias in the synthesis of this study. We have however included additional references based on our reply to comments from Reviewer 1 and 2. These are now in the manuscript bibliography (see also comment 2.8 from Reviewer 2). Additionally, we view that it is still important to consider other review papers in our manuscript to have a broad overview of the existing directions of the state of the ocean acidification sciences, and to draw parallels on whether the reviews highlighted similar knowledge gaps (see also comment 2.7 from Reviewer 2).
- Comment #1.3
I found the use of grammar in this manuscript often problematic. The references between sub-sentences are often unclear or possibly incorrectly constructed, which leads to misleading or even incomprehensible statements. A thorough linguistic revision would be necessary in any case. The usual reference to the involvement of a native speaker would not automatically be helpful in this case, because it would have to be a native speaker who could also grasp the meaning of the text in terms of content and check its correctness. I give some examples at the end.
Response: Thank you for your patience while reviewing our manuscript. We have proofread the manuscript via QuillBot Premium (https://quillbot.com/) using the Fluency function. The revision has been made accordingly throughout the manuscript.
- Comment #1.4
I would have expected from this review an overview on the carbon uptake mechanisms and carbon concentrating mechanisms in the different species that are listed in table 1 and I strongly suggest to complement the table with such information, which would be elemental to assess the sensitivity of different species to ocean acidification. I also found the first paragraph of section 3 - that is dedicated to this aspect - not very well structured. It would be better to (1) explain CO2 uptake by diffusion, then (2) explain the different CCMs and (3) address knowledge about their distribution among taxa.
Response: Thank you for the pointing this out. We have restructured this section to include the related information on CO2 uptake by diffusion, the different CCMs and addressing knowledge about their distribution of CCMs among the seaweed taxa. We also included the type of CCM for each species listed in Table 1. The related information is not updated in Table 1.
The amended text: line 262-276
- Comment #1.5
Some detailed points below, note that the list is not complete:
Title and throughout the text: I am not a native speaker, but I do not think the verb "to aquaculture" actually exists....
Response: We agree that the term ‘aquacultured seaweed’ is not commonly used. However, the same phrase had been applied in recently-published articles. E.g in: Liu et al., (2022) https://doi.org/10.1016/j.algal.2022.102654; Park et al. (2021) https://doi.org/10.1111/jwas.12786; and Kim et al., (2017) https://doi.org/10.4490/algae.2017.32.3.3 to name a few.
- Comment #1.6
- 14 may be reduced, not may reduce
Response: The revision has been made. It now reads:
Conversely, seaweed productivity may be reduced with ensuing economic losses. (line 17)
- Comment #1.7
l.36 what are "seaweed aquaculture beds (SABs) with a wide distribution of biomass globally"? Why would they (?) or their global distribution (?) have potential to act as a temporary carbon sink?
Response: Thank you for pointing this out. To clarify, examples of the global distribution of seaweed biomass has been mentioned in lines 49-56 and we have now included more details in Table 1. This statement is further explained by the following sentence with the supporting citations. It now reads:
Seaweed Aquaculture Beds (SABs) at the global level with a wide distribution of biomass have the potential to at least, act as a temporary carbon sink to mitigate the immediate effects of climate change [6]. This is due to seaweed's capacity for carbon assimilation, ac-cumulation, and CO2 sequestration in a relatively short period [4,7].
(line 41-45)
- Comment #1.8
- 47/48 Sentence fragmented and incomprehensive
Response: The sentence has been revised. It now reads:
Seaweeds are used in many different industries after they are harvested. As such, seaweed cultivation has grown from 45.9% of global mariculture production in 2004 to 51.3% in 2018.
(line 52-54)
- Comment #1.9
- 54-57 What is production of output of seaweeds? In any case, production should be singular, not plural...
Response: The sentence has been revised. It now reads:
The highest output of seaweeds by quantity farmed within temperate and tropical regions are: Eucheuma (also known as gusô, 10.2 million tons in 2015), Japanese kelp (8 million tons), Gracilaria (sare, 3.9 million tons), Undaria (wakame, 2.3 million tons), Kappaphycus (elkhorn sea moss, 1.8 million tons), and Porphyra (nori, 1.2 million tons) [1,12,16].
(line 60-64)
- Comment #1.10
L.58-59 Significant growth since when?
Response: We have clarified the sentence. It now reads:
From the temperate region, the seaweed production in Norway had reached a sales value of EUR 74,000 in 2017, while studies predicted the annual sales would rise further to EUR 4 billion by 2050 [17].
(line 64)
- Comment #1.11
- 61 not 95,6 % of all seaweed is "aquacultured", but 95,6% of all seaweed used by humans is "aquacultured" (see above about "to aquaculture")
Response: The sentence has been revised. It now reads:
Up to 95.6% of all seaweed used by humans is aquacultured to ensure sustainability in supply and to prevent overexploitation from the natural population [10].
(line 68-69)
- Comment #1.12
L63/64 Most brown seaweeds that are cultivated are actually cultivated for human nutrition, not for alginate production...
Response: We have amended this sentence and it now reads:
Brown seaweeds, which comprise nearly 64% of the farmed production, are harvested for variety of uses, including human nutrition, and alginate extraction.
(line 70-72)
- Comment #1.13
l.73/75 A negative effect of sea-level-rise on sea-based offshore aquaculture is not easy to understand, given that this takes usually place on floating infrastructructure, please explain...
Response: Thank you for pointing this out. To clarify, sea level rise and global warming are not likely to have a direct impact on floating cultures. Rather, they are more likely to have an impact on naturally growing seaweed that are harvested. The impacts on the floating cultures might not be obvious but changes of pH can still be considered as one of the factors which affect their physiological responses [27] (Table 1). We have now clarified the sentence and gist above had been incorporated in the revision. It now reads:
Rising sea-levels due to global warming may cause shifts in shoreline morphology with the subsequent effect of lowered productivity output through changes of seaweed distributional patterns [20]. Such changes may have more pronounced effects on naturally growing seaweed beds that would be harvested for seaweed industries rather than those in floating cultures [20]. Sea-level changes impacts on the floating cultures might not be obvious but changes of pH can still be considered as one of the factors which affect their physiological responses [27] (Table 1).
(line 79-85)
- Comment #1.14
- 94 & 223 Different carbon uptakes strategies should be explained in this paper, not just mentioned...
Response: Thank you for the suggestion. We have addressed this comment in response to comment 1.4 above in which we discussed about the carbon uptake strategies through different mechanisms. Particularly, we have added information about the carbon uptake process in CCM and and non-CCM species. For example, (i) obtaining CO2 solely through diffusive uptake, (ii) utilisation of DIC through active uptake to facilitate CO2 delivery, (iii) dehydration of HCO3- extracellularly. The gist is now in:
(line 262-276)
- Comment #1.15
- 228 why would CCM seaweeds have a low affinity to DIC...you probably rather mean non-CCM seaweeds?
Response: To clarify, according to Cornwall et al., 2017 [56], there is a difference between non-CCM and CCM with low affinity to DIC, which is also known as carbon-limited CCM species. The examples are Caulerpa prolifera and Dictyota dichotoma. We have now clarified the text to include these references:
- Cornwall, C. E.; Revill, A. T.; Hall-Spencer, J. M.; Milazzo, M.; Raven, J. A.; Hurd, C. L. inorganic carbon physiology underpins macroalgal responses to elevated CO2. Sci. Rep. 2017, 7, 1–12. https://doi.org/10.1038/srep46297.
- van der Loos, L. M.; Schmid, M.; Leal, P. P.; McGraw, C. M.; Britton, D.; Revill, A. T.; Virtue, P.; Nichols, P. D.; Hurd, C. L. Responses of macroalgae to CO2 enrichment cannot be inferred solely from their inorganic carbon uptake strategy. Ecol. Evol. 2019, 9, 125–140. https://doi.org/10.1002/ece3.4679
It now reads:
In contrast, CCM seaweeds with low affinity for DIC will benefit under elevated CO2 [140,143].
(line 275-276)
- Comment #1.16
L264/265 would certain seaweeds mitigate OA due to their high biomass or not rather their physiological ability, that permits them to build up high biomass?
Response: We agree that the physiological capacity of seaweeds to acclimatize in OA conditions is one of the factors that contribute to the production of high biomass. This factor is related to the intrinsic physiological characteristic of the seaweed species. Particularly, fleshy macroalgae consists of fast growing species and therefore would have a higher DIC uptake rate during photosynthesis. The revision has been made, it now reads:
Some seaweeds have the potential to mitigate OA through carbon sequestration [3,137,147] due to their respective physiological changes under pH variation thus leading to high biomass production [7], which was especially observed for fast-growing fleshy macroalgae (Table 1).
(line 293-296)
- Comment #1.17
- 269/270 Why would CCM species be particularly sensitive to low pH
Response: To clarify, based on the work by van der Loos et al., (2019), H+ is key in regulating cellular homeostasis. Changing H+ concentrations could affect macroalgal metabolic processes and CCM activity. For example (in Table 1), meiospore germination of Macrocystis pyrifera will be affected negatively under OA conditions. This is now revised in the text:
Decreased growth rates are also possible because CCM seaweeds are sensitive to increased H+ concentrations during OA, which can disrupt cellular homeostasis [140]. Studies of these effects are such as those on the changes of M. pyrifera meiospore germination (Table 1) under OA conditions at a mean pH of 7.60 [151].
(line 301-303)
- Comment #1.18
L 274 Do you mean the increase lasted for 16 h or photosynthesis alltogether lasted only for 16 h
Response: To clarify, the photosynthetic rate of the species did increased but only lasted for 16 h (range pH 6.5-8.0) and started to drop at pH 8.7. The text is now revised:
- Comment #1.19
- 345/347 Sentence fragmented
Response: We have clarified the sentence. It now reads:
The method of choice is primarily determined by the farm's scalable capacity and how to maximise productivity based on local environmental factors. [20].
(line 381-383)
- Comment #1.20
- 375 are more disease resistant
Response: We have clarified the sentence, but the revision is now structured to not be in need of the suggested text.
- Comment #1.21
L377 which means to
Response: We have clarified the sentence, but the revision is now structured to not be in need of the suggested text.
- Comment #1.22
L379 for a cultivation system
Response: We have clarified the sentence, "for cultivation system" has been replaced with "for a cultivation system"
- Comment #1.23
L383-385 Sentence fragmented
Response: We have clarified the sentence. It now reads:
Risk assessments based on an adaptive framework should be prioritised as part of the steps to address potential problems, such as threats from climate change. [8,176].
(line 420-421)
- Comment #1.24
L390 what are IUCN categories
Response: Thank you for pointing this out. Amendment made from "IUCN categories" to "International Union for Conservation of Nature (IUCN) categories".
(line 426-427)
- Comment #1.25
L391-393 Not clear
Response: The revision has been made. It now reads:
This is necessary to reduce the selection of farmed seaweed species that may exhibit negative growth rates in stressful and disturbed marine environments [56].
(line 422-424)
- Comment #1.26
L402 Seaweeds are the largest group of species in aquaculture in terms of what? Volume, species diversity, value?
Response: Thank you for pointing this out. We have now clarified the sentence. It now reads:
Seaweeds, as the largest group of aquacultured species with high productivity by volume, may be affected by regional and global changes in biomass yield.
(line 467-469)

Reviewer 2 Report
I have liked the idea of the work presented and it can be accepted for publication after they respond to my comments given below -
1. The data of seaweed production given is very old, the new data is now available with 2022 figures from FAO please incorporate it in the introductory section.
2. The background and purpose of the study need to be rewritten again, to make the objective clearer, just statement saying "knowledge gaps surrounding the impacts of OA on seaweed aquaculture" exist will not be sufficient.
3. Authors stated that they have searched relevant papers from Google Scholar and Scopus published between 2012 and 2021, what was rational to this time frame?
3. The meta analysis of the publications give some statistics on different types of research - but what is interesting to see how trends are changing over the period of time. I see authors have failed to bring out this aspects.
4. Figure 2 says about alginate and carrageenan, what about agar - this is also important polysaccharide, please include it.
5. I would like to see the inclusion of very concrete recommendations before the conclusion section.
6. There are good reviews on effect of Ocean Acidification on marine ecosystem and aquaculture as a whole, please try to bring parallels and adopt the ideas.
In short, authors needs to read a lot, before this paper is accepted.
Author Response
REVIEWER 2
- Comment #2.1
I have liked the idea of the work presented and it can be accepted for publication after they respond to my comments given below - The data of seaweed production given is very old, the new data is now available with 2022 figures from FAO please incorporate it in the introductory section.
Response: Thank you for the feedback and suggestion. The revision has been made, citing the suggested reference. It now reads:
This region led the world’s seaweed production with an estimated at 32.4 million tonnes in 2018, compared to a threefold lower production in 2000 (10.5 million tonnes; [14]). The global seaweed cultivation had then increased to 35.1 million tonnes in 2020 for food and non-food uses [15].
(line 57-59)
- Comment #2.2
The background and purpose of the study need to be rewritten again, to make the objective clearer, just statement saying "knowledge gaps surrounding the impacts of OA on seaweed aquaculture" exist will not be sufficient.
Response: Thank you for the comment. We have reorganised the flow of the existing text and have revised it accordingly. The revision is in lines 87-120
Knowledge on seaweed physiology, especially about how environmental stressors affect the productivity of aquacultured seaweed is clearly essential to ensure the success of seaweed farming [1].
(line 87-89)
In this regard, although studies regarding the impacts of OA to marine fauna or fisheries aquaculture have been conducted [38,41,42], there are still knowledge gaps surrounding the impacts of OA on seaweed aquaculture.
(line 103-106)
In this review, we discuss how the increase of dissolved CO2 with pH variation will affect the physiological responses of aquacultured seaweeds. In particular, we focused on directions to answer the following questions: (1) How do aquacultured seaweeds acclimatise with the increase of oceanic CO2? (2) What are the effects of OA on the photosynthetic rate and nutrient uptake of aquacultured seaweeds? And: (3) What are the knowledge gaps in the mitigation strategies for future seaweed aquaculture considering an ocean-acidified environment?
(line 114-120)
- Comment #2.3
Authors stated that they have searched relevant papers from Google Scholar and Scopus published between 2012 and 2021, what was rational to this time frame?
Response:
Thank you for the comment. We agree that before 2012 there were publications related to OA and seaweeds; however, the focus of this review is on the species itself and the emphasis on the timeframe for article selection would minimize duplication of the subject species, and to ensure that the most recent work in regards to the species of interest is provided. Furthermore, according to Snyder (2019)[48], this delimitation of the time frame is a methodological step to scoping a systematic review work. This has been mentioned in the Materials and Methods section. We did however checked papers before 2012, to ensure that any relevant papers were not overlooked. We have now clarified the text in this regard. It now reads:
The timeframe is consistent with the methodology for scoping systematic reviews [48] and specifically for this study, was stipulated to ensure that no species were overlooked. Furthermore, the emphasis on this timeframe for article selection was to minimise duplication of the subject species and to ensure that the published information provided was up to date. To avoid duplication of articles from the literature search, all articles obtained were imported into a reference manager (Mendeley), as suggested by Li et al. [49].
(line 148-154)
- Comment #2.4
The meta-analysis of the publications give some statistics on different types of research - but what is interesting to see how trends are changing over the period of time. I see authors have failed to bring out this aspects.
Response:
Thank you for the comment We would like to clarify that since we did not test a specific hypothesis, we view that statistical analysis is not necessary. Also, we assume that you are referring to the data in the Radar Chart (Fig 3). This chart aims to identify the ranking steps and to list out the topmost findings based on the number of hits from the different articles. The use of a radar chart enables the reader to maintain the focus on assessing the overall inherent issues and potential mitigation strategies for OA impacts on seaweed aquaculture. The visual presentation and holistic approach of a radar chart would then synthesise the findings of the proposed mitigation strategies while simultaneously making it easier to see ‘red flagged’ areas or areas where more focus is needed. Based on observation of the OA-related papers that are within the current scope, there appears to be more species studied, more investigations done on the multiple stressors on seaweed productivity, and more studies into species-specific CCM mechanism are focused. This observation is however not analysed quantitatively and is beyond the current scope of the research questions posed. There is room to incorporate this into future studies, which have been stated in Section 4.
- Comment #2.5
Figure 2 says about alginate and carrageenan, what about agar - this is also important polysaccharide, please include it.
Response: Thank you for pointing this out. The revision has been made in the updated Figure 2.
- Comment #2.6
I would like to see the inclusion of very concrete recommendations before the conclusion section.
Response: Thank you for the comment. The revision has been made in the Conclusion section. It now reads:
Seaweed aquaculture has the potential to reduce CO2 emissions while also supporting ecosystem services through CO2 sequestration; however, elevated CO2 and OA are likely to have an impact on seaweed production. To mitigate the negative effects of severe OA on aquacultured seaweed, a comprehensive mitigation plan with adequate monitoring is required. Seaweeds, as the largest group of aquacultured species with high productivity by volume, may be affected by regional and global changes in biomass yield. However, the responses of aquacultured seaweeds to OA vary by species, as evidenced by changes in physiological mechanisms such as photosynthetic rate and nutrient uptake, which affect seaweed productivity. At the same time, acclimatisation of aquacultured seaweed to elevated oceanic CO2 depends on their carbon uptake strategies, while kinetic mechanism adjustments would further determine changes in photosynthetic rate in each species. If OA has a negative effect on seaweed cultivation, the extent of its impact on seaweed production must be quantified. Because industry profitability is determined by seaweed price and operating expenses, monetary loss can be calculated precisely. More studies are needed to quantify the effects caused by OA on the economy. This includes establishing a link between physiological changes in seaweed and industrial productivity in terms of production costs and potential revenues. As a result, multiple mitigation strategies from various aspects should be implemented to mitigate the effects of OA on aquacultured seaweed. The emphasis should be on addressing existing knowledge gaps in mitigation approaches, which are still imbalanced and overly skewed toward monitoring- and IMTA-centric efforts. This entails multidisciplinary approaches developed through synergy among various stakeholders – from the researchers to aquaculture farmers and policymakers – for a more holistic seaweed aquaculture system that incorporates key mitigation tools. In short, the combined effects on biological and economic aspects necessitate the implementation of a more collaborative mitigation strategy that incorporates the various multidisciplinary aspects of OA and seaweed production.
(line 441-467)
- Comment #2.7
There are good reviews on effect of Ocean Acidification on marine ecosystem and aquaculture as a whole, please try to bring parallels and adopt the ideas.
Response: Thank you for the comment. The revision has been made based on the works that state the similarity in addressing OA on marine ecosystem and aquaculture as a whole, from the following review papers: Clement & Choppin 2016 [53]; Chung et al., 2017 [4]; Sondak et al., 2017 [7]
- Comment #2.8
In short, authors needs to read a lot, before this paper is accepted
Response: Thank you for your comment. After the revision made, additional references have been included. These are highlighted in the revised Reference section.

Round 2
Reviewer 2 Report
Authors have revised the manuscript by keeping in mind the comments I gave and the revised manuscript may be accepted for publication